# *SGCD* Missense Variant in a Lagotto Romagnolo Dog with Autosomal Recessively Inherited Limb-Girdle Muscular Dystrophy

**DOI:** 10.3390/genes14081641

**Published:** 2023-08-18

**Authors:** Barbara Brunetti, Barbara Bacci, Jessica Maria Abbate, Giorgia Tura, Orlando Paciello, Emanuela Vaccaro, Francesco Prisco, Gualtiero Gandini, Samuel Okonji, Andrea di Paola, Anna Letko, Cord Drögemüller, Vidhya Jagannathan, Maria Elena Turba, Tolulope Grace Ogundipe, Luca Lorenzini, Marco Rosati, Dimitra Psalla, Tosso Leeb, Michaela Drögemüller

**Affiliations:** 1Department of Veterinary Medical Sciences, University of Bologna, 40064 Bologna, Italy; barbara.bacci@unibo.it (B.B.); giorgia.tura@yahoo.it (G.T.); gualtiero.gandini@unibo.it (G.G.); samuel.okonji@unibo.it (S.O.); andrea.dipaola2@studio.unibo.it (A.d.P.); tolulope.ogundipe2@unibo.it (T.G.O.); luca.lorenzini8@unibo.it (L.L.); 2Department of Veterinary Sciences, University of Messina, 98168 Messina, Italy; jabbate@unime.it; 3Department of Veterinary Medicine and Animal Production, University of Naples Federico II, 80137 Naples, Italy; paciello@unina.it (O.P.); emanuela.vaccaro@unina.it (E.V.); francesco.prisco@unina.it (F.P.); 4Institute of Genetics, Vetsuisse Faculty, University of Bern, 3001 Bern, Switzerland; anna.letko@unibe.ch (A.L.); cord.droegemueller@unibe.ch (C.D.); vidhya.jagannathan@unibe.ch (V.J.); tosso.leeb@unibe.ch (T.L.); michaela.droegemueller@unibe.ch (M.D.); 5Genefast, 47122 Forlì, Italy; me.turba@genefast.com; 6Section of Clinical & Comparative Neuropathology, Centre for Clinical Veterinary Medicine, Ludwig-Maximilians-Universitaet-Muenchen, 80539 Munich, Germany; marco.rosati@outlook.de; 7Laboratory of Pathology, School of Veterinary Medicine, Faculty of Health Sciences, Aristotle University of Thessaloniki, 54124 Thessaloniki, Greece; dpsalla@vet.auth.gr

**Keywords:** sarcoglycan, *Canis lupus familiaris*, skeletal muscle, immunohistochemistry, precision medicine, Mendelian inheritance, development

## Abstract

An 8-month-old female Lagotto Romagnolo dog was presented for a 1-month history of an initial severe reluctance to move, rapidly progressing to a marked stiff gait and progressive muscular weakness and evolving to tetraparesis, which persuaded the owner to request euthanasia. A primary muscle pathology was supported by necropsy and histopathological findings. Macroscopically, the muscles were moderately atrophic, except for the diaphragm and the neck muscles, which were markedly thickened. Histologically, all the skeletal muscles examined showed atrophy, hypertrophy, necrosis with calcification of the fibers, and mild fibrosis and inflammation. On immunohistochemistry, all three dystrophin domains and sarcoglycan proteins were absent. On Western blot analysis, no band was present for *delta* sarcoglycan. We sequenced the genome of the affected dog and compared the data to more than 900 control genomes of different dog breeds. Genetic analysis revealed a homozygous private protein-changing variant in the *SGCD* gene encoding *delta-* sarcoglycan in the affected dog. The variant was predicted to induce a *SGCD*:p.(Leu242Pro) change in the protein. In silico tools predicted the change to be deleterious. Other 770 Lagotto Romagnolo dogs were genotyped for the variant and all found to be homozygous wild type. Based on current knowledge of gene function in other mammalian species, including humans, hamsters, and dogs, we propose the *SGCD* missense variant as the causative variant of the observed form of muscular dystrophy in the index case. The absence of the variant allele in the Lagotto Romagnolo breeding population indicates a rare allele that has appeared recently.

## 1. Introduction

The *SGCD* gene encodes the sarcolemmal transmembrane glycoprotein *delta*-sarcoglycan, one of four sarcoglycan proteins (*alpha-, beta-, gamma-*, and *delta*-SG) that are highly expressed in cardiac and skeletal muscle [1]. These four glycoproteins form the sarcoglycan complex, which in turn is part of the dystrophin–glycoprotein complex that stabilizes the sarcolemma. Disorders (also known as sarcoglycanopathies) resulting from mutated genes encoding the sarcoglycan–sarcospan complex [2] disrupt sarcolemmal integrity and cause limb-girdle muscular dystrophies (LGMDs) of types 2C–2F [3]. Limb-girdle muscular dystrophy is recognized as a genetic muscle disease with elevated serum creatine kinase levels and dystrophic changes in muscle histology [4]. Variants in the *DMD* gene encoding dystrophin cause the X-linked Duchenne and Becker muscular dystrophies, both of which are clinically similar to LGMD [5]. Once routine histology has confirmed dystrophic changes in the muscle biopsy, immunohistochemistry and immunoblot technology can be used to characterize specific protein deficiencies. Complementary genomic analyses allow the identification of the exact genetic etiology, and it is reported that pathogenic variants in any single sarcoglycan gene normally result in the absence of that protein and a secondary deficiency in all other sarcoglycan and dystrophin proteins [5,6].

Inherited diseases in domestic dogs have received considerable attention as a model for studying the genetics of human diseases, particularly as canine models provide access to experimental resources such as cells, tissues and even live animals for research and intervention purposes [7]. Sarcoglycanopathies have been described in several dog breeds: Chihuahua, Cocker Spaniel, and Boston Terrier [8,9]. In these first reports, the pathology was described from a clinical, histological, immunohistochemical and biochemical perspective. The cases described more recently in Boston Terriers (OMIA:002122-9615) and Miniature Dachshunds (OMIA:002305-9615) also describe the disease from a molecular point of view: two recessively inherited pathogenic *SGCD* variants [5] and one recessively inherited pathogenic *SGCA* variant [2] have been described. Both disease-causing *SGCD* loss-of-function variants were found to be homozygous in Boston Terrier dogs with limb-girdle muscular dystrophy type R6 (LGMDR6), specifically a two base-pair frameshift deletion in exon 6 and a frameshift deletion spanning exons 7 and 8 [5].

The aim of this paper is to describe a *delta* sarcoglycanopathy observed in a Lagotto Romagnolo dog from a clinical, histopathological, immunohistochemical, biochemical and molecular genetic point of view.

## 2. Materials and Methods

### 2.1. Clinical History

An 8-month-old female Lagotto Romagnolo dog was presented with a 1-month history of an initial severe reluctance to move, and on suspicion of a myopathy, the diagnostic workup included blood exams, urinalysis, PCR for neosporosis, toxoplasmosis and leishmaniosis, X-ray of the thorax, and electrodiagnostic tests associated with muscle and nerve biopsies. Biopsy specimens were obtained from the quadriceps femoris muscle, tibialis cranialis muscle and common peroneal nerve (frozen tissue).

Prednisolone (Prednicortone^®^ 5 mg, Dechra, Turin, Italy) was started at a dosage of 0.6 mg/kg twice a day without any clinical improvement.

Euthanasia was then performed at the owner’s request. The dog breeder was contacted to obtain further information on the dog’s parents and siblings and any blood samples.

### 2.2. Necropsy and Histology

During necropsy, tissue samples for histological evaluation were fixed in 10% buffered formalin for 24 h at room temperature, then embedded in paraffin and cut at a thickness of 4 microns. The sections were stained with hematoxylin and eosin (HE). Different muscles were sampled, including those that were macroscopically normal. Diaphragm, esophagus, tongue, thigh (quadriceps femoris), pectoralis major, right and left gluteus maximus muscle, sternocleidomastoid muscle, and myocardium were sampled.

### 2.3. Immunohistochemistry

Immunohistochemistry was performed using three antibodies labelling three different domains of the dystrophin protein. The rod domain was labelled with Dys1 antibody (raised against a peptide containing amino acids 1181–1388); the C-terminus domain was labelled with Dys2 antibody (raised against a peptide containing amino acids 3669–3685); and the N-terminus domain was labelled with Dys3 antibody (raised against a peptide containing the amino acids 321–494) [10]. Antibodies against *alpha-, beta-, gamm-a*, and *delta*-sarcoglycan and anti-laminin and spectrin were used (see Appendix A). Normal muscles of a dog of the same size from the archive of the University of Naples Federico II were used as control.

### 2.4. Western Blot Analysis

Snap-frozen muscle samples (quadriceps femoris muscle) from the affected dog and controls were cut at the cryostat at 20 μm and then lysed at 4 °C in 200 μL of TBS lysis buffer (Tris-buffered saline, 20 mM Tris-HCl pH 7.6, 140 mM NaCl, 30 mM sodium pyrophosphate, 5 mM ethylenediaminetetraacetic acid, 0.55% nonidet P40, 1% Triton X-100, 50 mM NaF, 0.1 mM Na_3_VO_4_, 1 mM phenylmethylsulfonyl fluoride, 1 mM benzamidine, 1 mM iodoacetamide, 1 mM phenanthroline). Protein concentration in the supernatant was determined by bicinchoninic acid (BCA) protein assay (BCA: Pierce Biotechnology, Rockford, IL), and lysates were adjusted to equivalent concentrations with lysis buffer. Aliquots of 10 mg of total muscle lysate were then separated on sodium dodecyl sulfate–polyacrylamide gel electrophoresis (SDS-PAGE). Proteins were transferred to polyvinylidene fluoride membranes that were blocked overnight at 4 °C with 5% nonfat dried skimmed milk in TTBS (TBS with 0.05% Tween 20). Incubation with primary specific antibodies against *delta*-sarcoglycan (ab115019) (1:500 dilution) and horseradish peroxidase-conjugated secondary antibodies was performed in blocking solution for 1 h at room temperature. Immunoreactive bands were visualized by a SuperSignal West Pico Chemiluminescent Substrate kit (Pierce Biotechnology, Rockford, IL). The same blots were stripped and reprobed using anti-GAPDH monoclonal antibody to confirm equal loading of proteins in all lanes.

### 2.5. Whole-Genome Sequencing (WGS)

An Illumina TruSeq PCR-free DNA library with an insert size of ~404 bp was generated from the affected female dog, using genomic DNA isolated from skeletal muscle. We collected 240 million 2 × 150 bp paired-end reads on a NovaSeq 6000 instrument (27× coverage). Mapping and alignment to the UU_Cfam_GSD_1.0 (GCF_011100685.1) reference genome assembly was performed as described [11].

### 2.6. Variant Calling

Variant calling was performed using GATK HaplotypeCaller [12] in gVCF mode as described in [11]. To predict the functional effects of the called variants, SnpEff software v5.0e [13,14] was used together with the NCBI annotation release 106 for the UU_Cfam_GSD_1.0 genome reference assembly. For variant filtering, we used 960 control dog genomes from genetically diverse different breeds, including 29 Lagotto Romagnolo dogs (see Appendix A). The integrative genomics viewer (IGV) software 2.8.2 [15] was used for visual inspection.

Numbering within the canine *SGCD* gene corresponds to the NCBI RefSeq accession numbers XM_038534930.1 (mRNA) and XP_038390858.1 (protein).

Run of homozygosity analysis was performed with detectRUNS v.0.9.6 [16] using the sliding window method. Based on previously published guidelines [17], values were set as follows: scanning window size of 50, max 5 missing and 3 heterozygous genotypes in a window, min run length of 300 kb, and min SNP density of 1 SNP/50 kb.

### 2.7. Genotyping Assays

Two genotyping tests were developed for the candidate variant *SGCD*:c.725T>C to confirm segregation with disease and to estimate the allele frequency in the population.

#### 2.7.1. PCR and Sanger Sequencing

On samples from the Vetsuisse Biobank, the candidate variant *SGCD*:c.725T>C was genotyped by direct Sanger sequencing of PCR amplicons. A 276 bp PCR product was amplified from genomic DNA using AmpliTaqGold360Mastermix (Thermo Fisher Scientific, Waltham, MA, USA) and the primers 5′-AAG TTA GAT GCT GCG AAA ATC C-3′ (forward) and 5′-GGT CTT TGA TAG GCT TTC TGT G-3′ (reverse). Sanger sequences were analyzed using the Sequencher 5.1 software (GeneCodes, Ann Arbor, MI, USA).

#### 2.7.2. Custom SNP Assay

On samples from a commercial veterinary laboratory (Genefast, Forlì-Cesena, Italy), the candidate variant *SGCD*:c.725T>C was genotyped by a Custom TaqMan™ SNP genotyping assay, non-human (assay ID ANKCM34, Thermo Fisher Scientific). Primers and probes sequences were, respectively: 5′-CTGTCTTTCCTTCCTATTCGGTTTCT-3′ (forward), 5′- CCTCGTTCCTGTAGGTGTGTAG-3′ (reverse); 5′-VIC TGCGAAAATCCAACTACCTAG MGBNFQ-3′ (probe 1 wild type); 5′-FAM CGAAAATCCAACCACCTAG MGBNFQ-3′ (probe 2 mutant). Taqman PCR was performed using TaqMan™ Universal Master Mix II, no UNG 1X (Thermo Fisher Scientific), Custom TaqMan SNP Assay 1X, DNA 2.5 µL and water to a final volume of 15 µL; PCR protocol was 95 °C for 10 min followed by 40 cycles of 95 °C for 15 s and 60 °C for 1 min.

## 3. Results

### 3.1. Clinical Description

After the initial severe reluctance to move, the dog rapidly progressing to a marked stiff gait. Dysphagia, dysphonia and polyuria and polydipsia appeared in the last five days prior to the examination.

General physical examination was unremarkable. Neurological examination showed a marked stiff gait (see Appendix A). Postural reaction and spinal reflex were normal. Dysphagia was detected. Muscle palpation elicited generalized severe pain.

The dog showed a progressive rapid worsening of the clinical signs leading in approximately one month to a severe non-ambulatory tetraparesis and severe dysphagia.

Blood exams showed biochemical signs of severe muscle damage (creatine kinase: 92,038 U/L, reference range 50–290 U/L; aspartate aminotransferase: 2290 U/L, reference range 15–52 U/L). Blood abnormalities included mild thrombocytosis (512,000, reference range 160,000–500,000), alanine aminotransferase increase (1506 U/L, reference range 15–65 U/L), and inflammation (C-reactive protein: 6.6 mg/dL, reference range 0–0.85 mg/dL). Urinalysis and X-ray of the thorax were normal.

Electromyographic examination revealed generalized abnormal muscle activity, characterized by fibrillation potentials and complex repetitive discharge (see Appendix A). PCR from muscle biopsies for neosporosis, toxoplasmosis and leishmaniosis were negative.

The dog breeder reported that the parents had died of intoxication and that the brothers were fine, but could not say where they were. Therefore, no related dogs were available for sampling.

### 3.2. Premortem Histology

Three muscle biopsies were obtained and showed similar histological features. Perimysium and endomysium presented with a normal amount of fibrocollagenous tissue. Myofiber density was normal. Myofiber diameters featured diffuse, moderate changes including numerous small polygonal atrophic fibers intermingled with multiple hypertrophic round non-lobulated myofibers. Multiple fibers featured polyphasic myonecrosis and myophagocytosis accompanied by mild, perilesional, interstitial lymphohistiocytic infiltrates. Nuclear internalization and occasional myofiber splitting were also observed. Multiple fibers undergoing dystrophic mineralization were detected in all sections.

The diagnosis was a necrotizing myopathy/myositis, polyphasic, diffuse, chronic/active and moderate with dystrophic mineralization of myofibers.

Semithin sections of the nerve were available, which displayed a diffuse, moderate reduction in nerve fiber density accompanied by a moderately increased amount of endoneurial fibrocollagenous tissue and edema. Axonal diameters appeared normal. Myelin sheaths showed multiple hypomyelinated fibers and rare myelin ovoids. Occasionally, hypertrophic Schwann cells were also seen.

Further evidence of mild neuropathy with endoneurial mononuclear cell infiltration was found.

### 3.3. Necropsy Examination

At necropsy, the dog was moderately emaciated. The only muscles that appeared grossly hypertrophic were the diaphragm (diffuse thickness of 1 cm, Figure 1A) and the muscles of the neck. The other muscles showed moderate atrophy. The heart showed moderate mitral valve endocardiosis (Figure 1B).

### 3.4. Postmortem Histology

All muscles examined showed the typical dystrophic phenotype, and the severity of the injuries was variable among the different muscles. Lesions were severe in the diaphragm only, moderate in the esophagus, tongue, and neck muscles, and mild in all other muscles examined. The cardiac muscle was histologically normal. On HE sections, all skeletal muscles (except for the heart muscle) were characterized by moderate variation in myofiber diameter ranging from atrophic to hypertrophic myofibers that occasionally exhibited splitting, and multifocal polyphasic necrotic changes with occasional myofiber mineralization (dystrophic myofiber mineralization). Most muscle fibers showed pyknotic nuclei, intensely eosinophilic sarcoplasm, loss of cross striations and intact basal lamina (segmental necrosis). Multifocally, small clusters of necrotic myofibers were infiltrated by moderate numbers of macrophages (myophagocytosis). Mild muscle regeneration was observed, with fiber splitting and scattered muscle showing a small basophilic sarcoplasm with multiple, linearly arranged central active nuclei. Mild multifocal endomysial and perimysial fibrosis and infiltration of histiocytes and lymphocytes were observed (Figure 2). Finally, mild adipose tissue replacement was observed in all examined muscles and was more prominent in the esophageal musculature and tongue. A chronic, multifocal, polyphasic necrotizing myopathy with dystrophic myofiber mineralization, muscle regeneration, fibrosis and adipose tissue replacement was diagnosed. Histological findings confirmed the premortem suspicion of a primary muscular degenerative disease. Intramuscular nerve branches within investigated skeletal muscles were within normal limits.

### 3.5. Immunohistochemical Results

Three different antibodies labelling three different domains of the dystrophin (Dys1: rod-domain, Dys2: C-terminus domain, Dys3: N-terminus domain) were used to assess the presence of even truncated forms of this protein. Immunohistochemistry revealed the loss of expression of all three dystrophin domains on the sarcolemma of almost all muscle fibers in the diaphragm, with a few fibers showing partial or full membrane positivity (revertant fibers). In the myocardium, there was a diffuse loss of expression of the rod and C-terminus domain (Dys1 and Dys2), but nearly normal expression of the N-terminus domain (Dys3). Immunohistochemistry for *delta*-sarcoglycan showed a total absence of expression in the diaphragm muscle and heart (Figure 3). The other three types of sarcoglycan were tested only on frozen muscles, but equally there was largely absence of expression, with the presence of some revertant fibers (see Appendix A). The expression of laminin and spectrin also tested on frozen muscle was normal.

### 3.6. Western Blot Analysis

Sample 1 from the presented dog showed no band for the *delta*-sarcoglycan (Figure 4).

### 3.7. Identification of a Candidate Causal Genetic Variant

We sequenced the genome of the affected dog and searched for private protein-changing variants that were exclusively present in either heterozygous or homozygous state and absent or only heterozygous in the genomes of 960 other dogs, including 29 Lagotto Romagnolo dogs.

We considered two alternative scenarios for the putative causal variant: for an autosomal recessive trait, we expected the affected dog to be homozygous for the alternative allele and controls to be homozygous for the reference allele or heterozygous. Conversely, for a dominant trait that could only have been caused by a de novo mutation event, the affected dog should be heterozygous and all controls should be homozygous for the reference allele.

The variant filtering for a possibly dominant acting de novo mutation revealed 15 heterozygous protein-changing variants (see Appendix A), but no obvious candidate gene was involved.

The analysis identified 10 homozygous protein-changing variants affecting eight different annotated genes and two uncharacterized loci (see Appendix A). We prioritized a single homozygous private protein-changing variant in exon 8 of the *SGCD* gene, a known candidate gene for recessively inherited forms of muscular dystrophy (OMIM 601287). The other nine private protein-changing variants were not located in genes known to cause similar phenotypes in humans, mice, or domestic animals. The *SGCD* variant can be designated chr4:54,154,870A>G (UU_Cfam_GSD_1.0 assembly) (Figure 5). It is a missense variant, XM_038534930.1: c.725T>C, predicted to change a highly conserved leucine residue in the C-terminal extracellular domain of *SGCD*, XP_038390858.1:p.(Leu242Pro). In silico analysis predicted the functional effect of p.(Leu242Pro) as deleterious using PredictSNP 1.0 software [13,14]. The affected dog had an 11.7 Mb region of homozygosity on chromosome 4 from 43.3 to 55.0 Mb.

We confirmed the presence of the *SGCD* missense variant by Sanger sequencing (Figure 5b). The affected dog carried the mutant allele in a homozygous state (Table 1). We also genotyped the *SGCD*: c.725T>C variant in a population control cohort comprising 741 Lagotto Romagnolo dogs without any phenotypic records. The mutant *SGCD* allele was detected only in the affected dog, whereas it was absent in all controls (Table 1).

## 4. Discussion

Limb-girdle muscular dystrophies (LGMD) are a heterogeneous group of genetic diseases that in humans are characterized by progressive muscle wasting that affects predominantly hip and shoulder muscles [18]. LGMD type 2F (LGMD2F) is caused by mutations in the *SGCD* gene, which encodes for the protein *delta*-sarcoglycan [18]. This protein is part of a complex of proteins that comprise the dystrophin–glycoprotein complex (DGC), which plays an important role in maintaining muscle cell integrity. Deficiency in *delta*-sarcoglycan leads to a loss of DGC function, which in turn leads to muscle cell damage and necrosis [19].

The overall pattern of pathology is usually dystrophic and is indistinguishable from Duchenne and Becker muscular dystrophies; therefore, molecular analyses are mandatory for a definitive diagnosis [5,6]. Based on the presented clinicopathological and genetic data, an inherited form of limb-girdle muscular dystrophy (*delta* sarcoglycanopathy) characterized by severe clinical, biochemical and histopathological signs of muscle damage was diagnosed in a single Lagotto Romagnolo dog homozygous for the *SGCD* missense variant. Initially, we hypothesized that a rare breed-specific deleterious variant was responsible for the diagnosed disease phenotype in our case. Homozygosity mapping revealed an extended interval of identity by descent flanking the most likely pathogenic variant in the *SGCD* gene on chromosome 4, supporting the putative recessive mode of inheritance that was described for similar forms of muscular dystrophy in dogs and other species. No heterozygous carriers of the variant were found among hundreds of unrelated normal dogs or Lagotto Romagnolo dogs. This suggests a very rare variant, possibly of recent origin. The run of homozygosity analysis revealed an autosomal average genomic inbreeding (F_ROH_) value of 0.426 for the sequenced case. This is very high compared to the other 29 sequenced dogs in the breed, in which the average F_ROH_ score was 0.263. This supports the assumption already made that the affected dog most likely originated from an (unfortunately undocumented) inbred mating. The disease-associated 11.7 Mb haplotype was not found in any of the other 29 available genome sequences from unaffected Lagotto Romagnolo dogs. This observation is consistent with the apparent rarity of the *SGCD* allele in the studied population.

In humans, various forms of recessively inherited limb-girdle muscular dystrophy type 6 are caused by missense, nonsense or frameshift variants affecting *SGCD* (OMIM:601287). In dogs, two independent pathogenic *SGCD* variants in Boston Terriers have been reported to cause a very similar form of the disease (OMIA:002122-9615) [4]. Molecular genetic testing is commonly used to identify pathogenic variants in the human sarcoglycan genes, obvious functional candidates for the more severe forms of LGMD [2]. The WGS of a single affected dog was used to identify the most likely pathogenic *SGCD* variant (p.(Leu242Pro)), which was absent in all other control dogs tested. This amino acid exchange affects a residue highly conserved across multiple species. The literature as well as human genome databases, such as GnomAD or ClinVar, do not provide evidence for the presence of variation at the corresponding human position [20]. Therefore, we report for the first time a variant affecting this residue that appears to be associated with a disease phenotype similar to those previously described in Boston Terriers homozygous for loss-of-function frameshift variants. Based on current knowledge of gene function in other mammalian species, we propose the *SGCD* missense variant as the most likely causative variant for the observed form of muscular dystrophy in the presented dog.

From a pathological point of view, it is interesting to note that although all muscles except cardiac muscles were microscopically affected by the lesions to some degree, the diaphragm and the neck muscles were the only muscles macroscopically affected by thickening.

As already reported in the literature [8], a mutation of *delta*-sarcoglycan can alter the expression of dystrophin. In fact, in the present case, despite dystrophin not being mutated, its expression was lost at the protein/immunohistochemical level as a consequence of the *delta*-sarcoglycan defect. Therefore, not only a deletion of *delta*-sarcoglycan but a missense variant may prevent the expression of the other sarcoglycans and dystrophin. It is also of interest that the heart muscle was histologically unremarkable. In human literature, dilatated cardiomyopathy is associated with mutation of dystrophin and tafazzin, cardiac actin, desmin and laminin A/C and also the sarcoglycans, in particular *delta*-sarcoglycan [21]. However, the cardiac muscles showed immunohistochemical changes that consisted of loss of expression of *delta*-sarcoglycan and partly of the dystrophin protein, possibly indicating a dysfunction of the dystrophin–glycoprotein complex. These results suggest that the lack of *delta*-sarcoglycan and of a part of dystrophin may have different effects on the myocardium compared to other muscle groups.

## 5. Conclusions

In conclusion, we have identified a non-synonymous variant in a highly plausible functional candidate gene using WGS data analysis and demonstrated how a missense mutation in *SGCD* can alter the protein/immunohistochemical expression of other sarcoglycans and dystrophin, resulting in severe muscle pathology. Our results, combined with current knowledge of *SGCD* function in other species, provide strong evidence for an extremely rare missense variant affecting a conserved residue of *SGCD* as the most likely causative genetic variant for recessive limb-girdle muscular dystrophy in the Lagotto Romagnolo dog. This is the second report on the underlying pathogenesis of LGMD in dogs, confirming the efficacy of the method chosen for gene discovery in rare muscular diseases and enabling genetic testing for veterinary diagnostic and breeding purposes.

## Figures and Tables

**Figure 1 genes-14-01641-f001:**
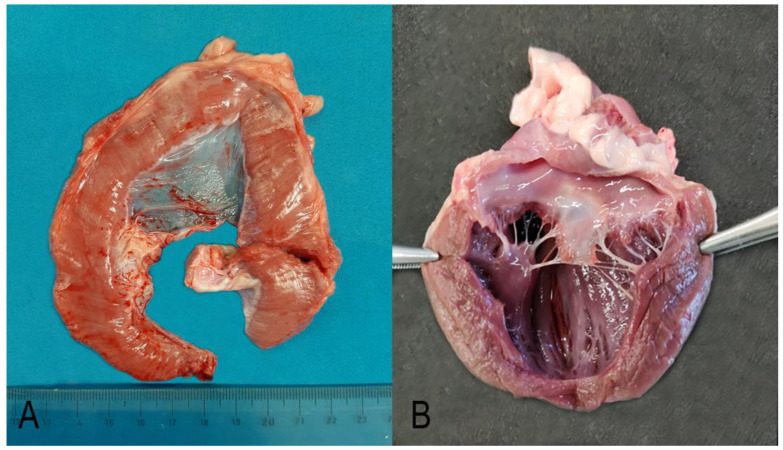
Diaphragm (**A**). The diaphragmatic muscle was diffusely thickened up to 1 cm and showed several white multifocal stripes parallel to the muscle fibers. Heart (**B**). The mitral valve cusps were diffusely thick and showed round margins and prominent nodular thickenings and retraction on the free margin (endocardiosis).

**Figure 2 genes-14-01641-f002:**
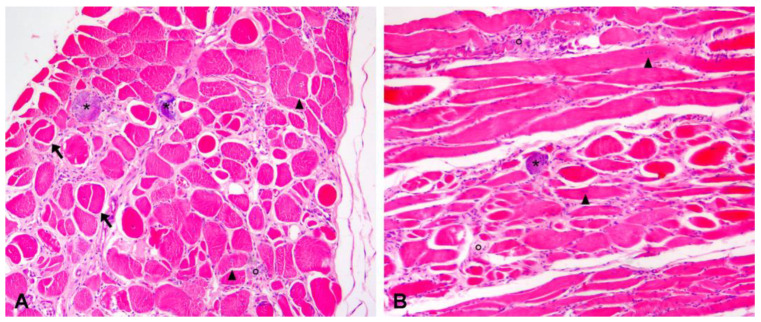
Hematoxylin and eosin-stained transversal (**A**) and longitudinal (**B**) histological sections of the diaphragm muscle (10× magnification). (**A**) In the transversal section, there is a moderate variation in myofiber diameter, ranging from atrophic to hypertrophic myofibers, and multifocal widening of the endomysium due to fibrosis and infiltration of inflammatory cells. Nuclear internalization (arrowheads), fiber splitting (arrows), and multifocal myofiber necrosis (°) with mineralization (*) are also evident. (**B**) In the longitudinal section, it is possible to appreciate the arrangement in rows of plump internalized nuclei (arrowheads) and myofiber necrosis (°) with occasional mineralization (*).

**Figure 3 genes-14-01641-f003:**
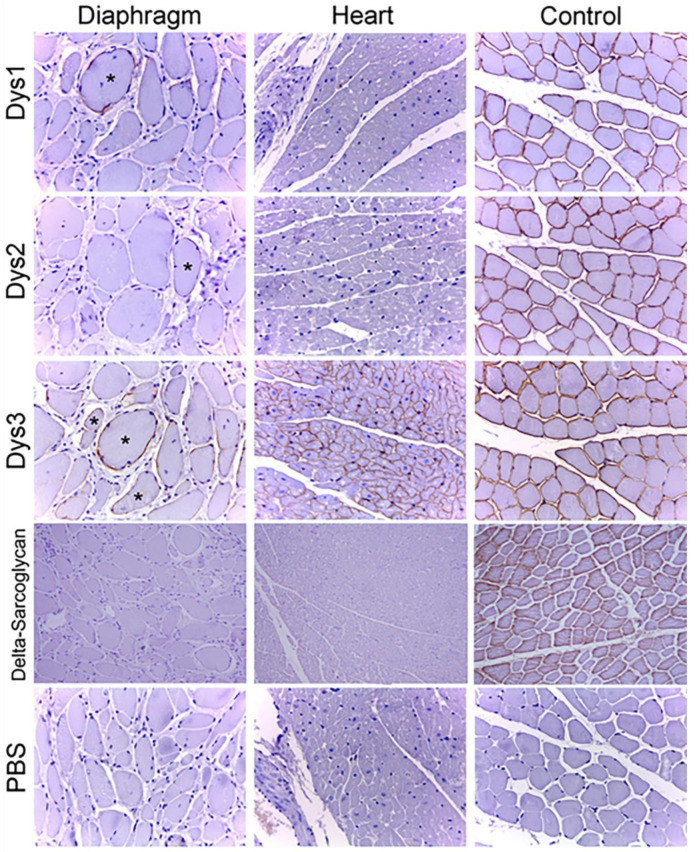
Immunohistochemistry for dystrophin rod (Dys1), C-terminus (Dys2), N-terminus (Dys3) domains and *delta*-sarcoglycan on section of diaphragm muscle and heart (20× magnification). Note the absence of expression of dystrophin with the exception of occasional revertant fibers (*) and the absence of expression of *delta*-sarcoglycan in diaphragm muscle and heart.

**Figure 4 genes-14-01641-f004:**
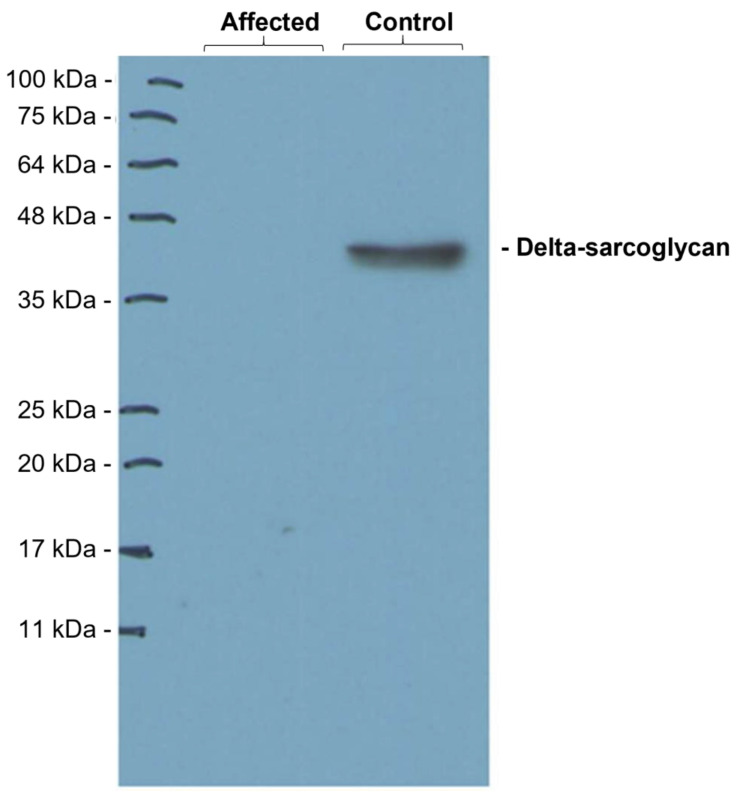
Representative image of the Western blotting analyses performed to detect *delta*-sarcoglycan in the muscle samples from the affected and a control dog. Only the control dogs show a band corresponding to *delta*-sarcoglycan.

**Figure 5 genes-14-01641-f005:**
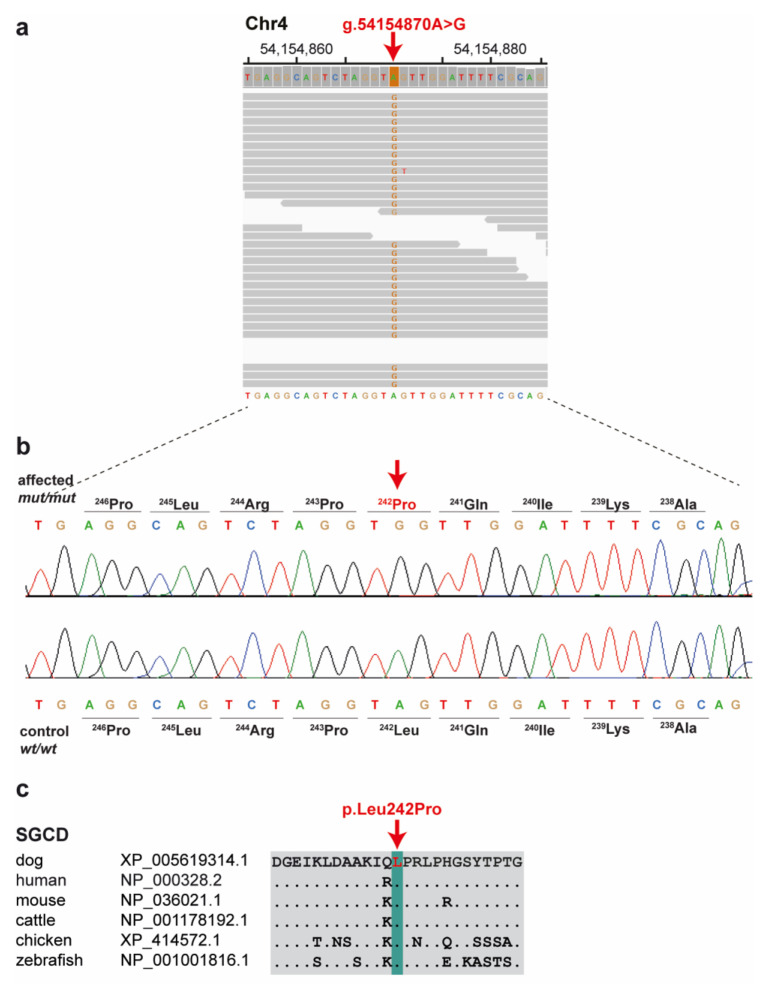
A missense variant in the *delta*-sarcoglycan (*SGCD*) gene is associated with muscular dystrophy in a Lagotto Romagnolo dog. (**a**) IGV [15] screenshot of the short-read alignments from the affected dog shows the missense variant in exon 8. (**b**) Sanger electropherograms of the case and an unaffected control dog indicate the c.725T>C variant that changes codon 242. (**c**) Evolutionary conservation of the SGCD protein across multiple species. Leucine-242 is perfectly conserved in diverse vertebrate species.

**Table 1 genes-14-01641-t001:** Association of the missense variant in *SGCD*:c.725T>C with the muscular dystrophy phenotype in Lagotto Romagnolo dogs.

	Ref/Ref	Ref/Alt	Alt/Alt
Muscular dystrophy-affected dog			1
Unrelated Lagotto Romagnolo dogs from Italy ^a^	293		
Other Lagotto Romagnolo dogs ^a^	448		
Sequenced Lagotto Romagnolo genomes ^b^	29		
Sequenced dog genomes from various other breeds	931		

^a^ Phenotypes are unknown. ^b^ DBVDC cohort [11].

## Data Availability

The sequence data of the sequenced dog was deposited in the European Nucleotide Archive under study accession number PRJEB16012 and sample accession number SAMEA110415697.

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
