# Peer review of "SGCD Missense Variant in a Lagotto Romagnolo Dog with Autosomal Recessively Inherited Limb-Girdle Muscular Dystrophy"

_genes, 2023, doi:10.3390/genes14081641_

Round 1
Reviewer 1 Report
The manuscript ‘SGCD missense variant in a Lagotto Romagnolo dog with recessively inherited limb-girdle muscular dystrophy’ describes the pathology and molecular genetic study of a single case. It is well written; the methodology is straightforward and the conclusions are sound. I do have a number of minor concerns.
1. The source of the DNA of the dog is not mentioned. When discussing possible de novo DNA mutations, the source is not irrelevant.
2. It should be described what is known of the ancestry of the dog. Was it a pedigree dog from the Italian population or a look-alike? Did the parents have common ancestors? Couldn’t any related dogs be traced? It is highly unlikely that a variant with an allele frequency of less than 0.001 pops up homozygously, unless there is close inbreeding. What is the overall level of homozygosity of the dog and how does it compare with other dogs of the breed that have been sequenced?
3. It is suggested that the SGCD mutation could be recent, apparently because it has not spread in the population (line 341). Therefore, it is relevant to establish from available WGS data whether the haplotype surrounding the mutation is observed in other dogs of the breed.
4. Line 38: It is unusual to call a homozygous dog a carrier. Usually, carriers are heterozygotes for a recessive allele.
5. In my opinion SGCD in the title should be in italics. I suggest removing ‘recessively inherited’ from the title because it implies some kind of foreknowledge.
6. Typo in Table 1: ‘unrekated’
7. Line 318: Consider replacing ‘all vertebrate species’ by ‘diverse vertebrate species’.
8. Without knowledge of the ancestry of the dog, it should not be called ‘purebred’ and the inbreeding should not be called ‘accidental’ (lines 335 and 342).
9. Line 386: Genetic testing for breeding purposes is mentioned. In the light of the apparent absence of the variant in the general population this is uncalled for. The stated ‘no conflict of interest’ is not correct because one of the co-authors is employed by an animal DNA testing company.
Author Response
-

Reviewer 2 Report
This is a well written paper characterizing a novel neurological disease in Lagotto breed and describing a novel variant/mutation causing limb-girdle muscular dystrophy. All of the appropriate studies were also performed, supporting that the discovered variant is indeed the disease causing variant. I only have two comments: I would add "autosomal" before recessively in the title to make it very clear that it is not like DMD, which is X-linked recessive. The other was a typo in Table 1: It should say unrelated instead of unrekated. It was a pleasure to review this paper.
Author Response
-
